

# Ultra-high energy neutrinos searches with the Pierre Auger Observatory

**Marta Trini[1] on behalf of the Pierre Auger Collaboration[2]⋆**

**1** Center for Astrophysics and Cosmology (CAC),
University of Nova Gorica, Nova Gorica, Slovenia
**2** Observatorio Pierre Auger, Av. San Martín Norte 304, 5613 Malargüe, Argentina

⋆ auger_spokespersons@fnal.gov

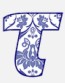

## Abstract

**In the EeV range, neutrinos are expected to be produced by ultra-high energy cosmic rays interactions with the Cosmic Microwave Background during propagation in the Universe. We report on the search for ultra-high energy neutrinos in data collected with the Surface Detector of the Pierre Auger Observatory. The searches are most efficient in the zenith angle range from 60 degrees to 95 degrees with tau neutrinos skimming in the Earth playing a dominant role. The present non-detection of UHE neutrinos in the Pierre Auger Observatory excludes the most optimistic scenarios of neutrino production in terms of UHE cosmic rays chemical composition and cosmological evolution of the acceleration sites. We also report on the searches for neutrinos in coincidence with the recent Gravitational Wave events detected by LIGO/Virgo.**

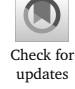

## 1 Introduction

The observation of ultra-high energy cosmic rays (UHECRs) of energies above $10^{17}$ eV, implies the production of ultra-high energy neutrinos (UHE$\nu$s). Above $\sim5\times10^{19}$eV cosmic rays, composed of protons and heavier nuclei, interact with Cosmic Microwave Background photons and produce cosmogenic UHE$\nu$s. UHE$\nu$s are also expected to be produced from the decay of charged pions created in the interactions of $>10^{17}$ eV accelerated hadrons with surrounding matter and radiation at the source. The detection of ultra-high energy neutrinos could provide unique information about their origin since, unlike cosmic rays, they point directly to the source and travel unabsorbed along cosmological distances. Ultra-high energy neutrinos can be indirectly observed by cosmic-ray particle shower detectors such as the Pierre Auger Observatory.

## 2 Ultra-high energy neutrinos at the Pierre Auger Observatory

The Pierre Auger Observatory, located in Malargüe, Argentina, is the largest existing hybrid (employing two independent methods of data collection) air shower detector. It consists of an array of surface particle detectors (SD) on 3000 km$^2$ and four fluorescence telescope sites [1]. The SD array consists of about 1600 purified water Cherenkov detectors arranged in a triangular grid with 1.5 km spacing between them. As charged particles enter in a SD station they emit Cherenkov light which is collected by photomultiplier tubes and converted to a digital signal with a temporal resolution of 25 ns. The identification of neutrino showers with the SD of the Auger Observatory can be performed efficiently as long as the search is restricted to showers arriving with high zenith angle ($\theta$). The electromagnetic component (EM) of the inclined hadronic showers, which initiate high in the atmosphere, will be absorbed before reaching the ground-level. Neutrino showers with the first interaction happening deep in the atmosphere have a significantly large EM component at ground. As a consequence, neutrino-induced showers will produce a signal spread in time over hundreds of nanoseconds, while hadronic showers at ground are muon-dominated and will induce nanosecond signals. Muons, in fact, travel in a straight line following the shower axis while photons and electrons are affected by multiple scattering in the atmosphere and they arrive with a larger spread in time. Due to this feature, the large background of hadronic showers can be separated from the neutrino-induced showers. Downward-going neutrinos can interact through charged-current (CC) or neutral-current (NC) with air nuclei. As evinced from Monte Carlo simulations, this search can be performed more efficiently as long as it is restricted to deep showers with high zenith angles divided in two angular subranges: downward-going low (DGL), $60° < \theta < 75°$, and downward-going high (DGH), $75° < \theta < 90°$. Moreover horizontal EeV tau neutrinos skimming the Earth's surface can interact via CC and produce a tau lepton which can emerge in the atmosphere and decay in flight producing detectable up-going showers. Typically, only Earth-skimming (ES) tau neutrino with zenith angles $90° < \theta < 95°$ may be identified. The ES channel increases the possibility of detecting tau neutrinos, making the Pierre Auger Observatory one of the most sensitive neutrino detectors in the EeV range [2].

With the SD of the Auger Observatory we are sensitive to neutrinos in a broad declination ($\delta$) range spanning $\Delta\delta \sim 150°$, with the sensitivity (proportional to the fraction of time a source is seen) peaking at equatorial declinations $\delta = 55°$ and $\delta = -55°$. The surroundings of the Northern Terrestrial Pole ($60° < \delta < 90°$) is a blind region in this analysis.

The selection procedure of inclined and deep showers requires that: the triggered SD stations exhibit an elongated pattern on the ground, the apparent speed of signal must be close to speed of light, compatible with an event traveling in nearly horizontal direction and the signal has to be broad in time to be induced by the electromagnetic instead of the muonic component.

These criteria were applied blindly to search for ES, DGL and DGH neutrinos in the data collected with the SD from 1 January 2004 to 31 March 2017. No neutrino candidate was found. As a result a limit to the diffuse neutrino flux as well as a function of declination can be obtained [3] (Fig. 1). From the current limits it is possible to put constraints on the composition of UHECRs at the sources and to how strongly their comoving density evolves with redshift.

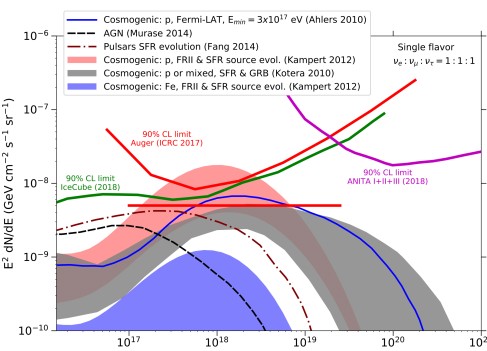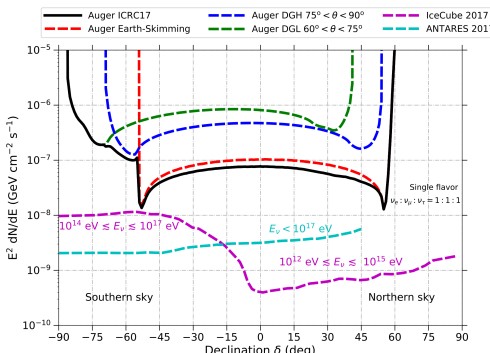

Figure 1: Left: Integral upper limit (at 90% C.L.) for the normalization, k, of a diffuse flux $dN/dE = kE^{-2}$ of single flavor neutrinos and differential upper limit compared to other experiment and several model predictions. All flavors are assumed to have equal fluxes. Right: Upper limits at 90% C.L. for a single flavor neutrino flux as a function of declination, $\delta$, for ES, DG channels, compared to other experiments.

## 3 Ultra-high energy neutrino follow-up of Gravitational Wave events with the Pierre Auger Observatory

During 2015, the two Advanced LIGO detectors observed the first gravitational wave (GW) transients: GW150914, GW151226 and a candidate event, LVT151012 [4]. The GW events were inferred to arise from mergers of a black-hole binary systems. Binary black-holes mergers are not supposed to produce an electromagnetic or neutrino counterpart: gas around binary black-holes would have been swept long before their final encounter. However such systems may still accelerate ultra-high energy cosmic rays, which can produce ultra-high energy neutrinos, as long as there are relic magnetic fields and debris from the formation of the two black-holes [5].

The Surface Detector array of the Pierre Auger Observatory can detect air showers induced by ultra-high neutrinos from the inferred position in the sky of the GW events. Unfortunately the 90% CL position contour for event GW150914 and the Auger field of view did not overlap at the time of the merger. However a significant portion of the position of the source was visible for a fraction of time in 1 day after the corresponding GW event, as the Earth rotates and the fields of view of the ES and the DGH analyses move through the sky. In the case of GW151226 there is a significant overlap at the time of the merger and in 1 day after the corresponding merger event. A targeted search for highly inclined and deep air showers yielded no candidates for the Auger data collected ± 500 s around the UTC time of the GW events and in the period of 1 day after [6].

These specific time windows of the searches derive from the association of these mergers of compact systems to short Gamma-ray bursts (GRBs). Given the fact that no neutrino candidates have been identified, a limit to the neutrino fluence, i.e. the energy radiated in ultra-high energy neutrinos per unit area, can be obtained combining ES and DGH channels (Fig. 2). The constraints obtained from the non-observation of ultra-high energy neutrinos can be used to construct stellar black-hole merger models. Stellar black-hole mergers can accelerate UHECRs (as long as there is a relic magnetic field) to energies $\sim 10^{19}$ eV with <3% of the energy released into particle acceleration. Auger limits are less restrictive than IceCube/ANTARES ones but they apply in complementary energy ranges ($E_\nu < 10^{17}$ eV for IceCube/ANTARES vs $E_\nu > 10^{17}$ eV for Auger).

In August 2017 the Advanced LIGO and Advanced Virgo experiments detected another

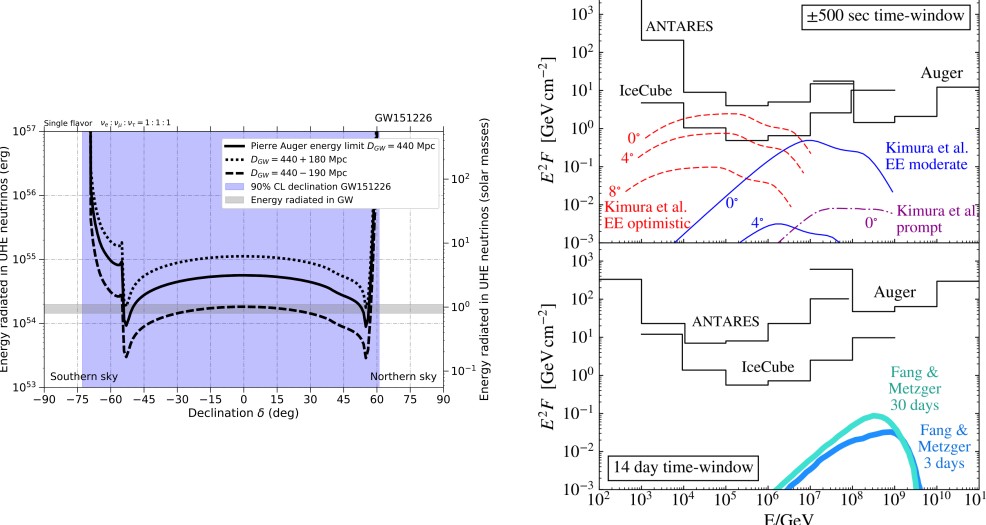

Figure 2: Left: Upper limit to the energy radiated in ultra-high energy neutrinos from GW151226 (Binary Black Hole merger) as a function of equatorial declination [9]. The limits only apply in the blue band. Right: Upper limit to the ultra-high energy neutrino fluence per flavor from GW170817 (Binary Neutron Star merger).

pulse of gravitational waves, known as GW170817 [7]. A short gamma-ray burst following the event was observed by the Gamma-ray Burst Monitor (GBM) on NASA's Fermi Gamma-ray Space Telescope. As the GRB direction was compatible with the GW170817 position, it was associated with a binary neutron star merger.

Subsequent optical observations allowed the localization of the merger in the galaxy NGC 4993 (at about 40 Mpc). The Pierre Auger Observatory together with dedicated neutrino experiments ANTARES and IceCube searched for high-energy neutrinos correlated with this event [8] (Fig. 2).

No neutrino candidate directionally coincident with the merger was found within 500 s or within 14-day period following the merger. This non-detection is consistent with model predictions of a short GRB observed off-axis.

## 4 Conclusions

The analysis of the data collected with the Pierre Auger Observatory from 1 January 2004 to 31 March 2017 revealed no candidate events that allowed us to put very sensitive bounds on neutrino flux at $10^{18}$ eV. The Pierre Auger Observatory search for UHE neutrinos, in an energy range between 100 PeV and 25 EeV is complementary to those of IceCube/ANTARES that apply in the energy range between 100 GeV and 100 PeV. Upper limits to the flux of ultra-high energy neutrinos can be placed for a diffuse flux (integral and differential) and as a function of declination as well as from the sources of GW events. With the ES channel there is an enhanced probability of detecting tau neutrinos, making the Pierre Auger Observatory one of the most sensitive detectors in the EeV range. The recent detection of two upward-going events by the ANITA detector [10] if interpreted as showers induced by tau leptons of tau neutrino origin, would imply a flux of tau neutrinos that should have already been detected with great significance with the Pierre Auger Observatory [11].

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
