# Peer review of "Ultra-high energy neutrinos searches with the Pierre Auger Observatory"

_SciPost Physics Proceedings, doi:SciPost Phys. Proc. 1, 047 (2019)_

## Round 1 · Referee Report · Nicolo De Groot · 2018-11-15

Strengths

1. Very readable article
2. Covering an interesting, timely topic

Weaknesses

no significant weaknesses

Report

This is a nice conference paper about the neutrino results from Auger. I recommend publishing it.

Requested changes

1. p1, Introduction: The interactions with CMB start at 5x10^19 eV, so why UHECRs of 10^17 eV imply the production of UHE neutrinos ? Please clarify

2. p2: Explain why does the EM component of neutrino showers lead to such a large signal spread

3. p2: please refer to http://auger.org/education/Auger_Education/celestialcoordinates.html for the benefit of non-experts on Auger coordinates

4. Any comments on the Anita HE upwards neutrino events ?

---

## Editorial Decision

published